# Germination response of diverse wild and landrace chile peppers (*Capsicum* spp.) under drought stress simulated with polyethylene glycol

**Vivian M. Bernau**[1¤], **Lev Jardón Barbolla**[2], **Leah K. McHale**[1], **Kristin L. Mercer**[1]*

**1** Department of Horticulture and Crop Science, The Ohio State University, Columbus, Ohio, United States of America, **2** Centro de Investigaciones Interdisciplinarias en Ciencias y Humanidades, Universidad Nacional Autónoma de México, Mexico City, Mexico

¤ Current address: North Central Region Plant Introduction Station, Agriculture Research Service, United States Department of Agriculture and Department of Agronomy, Iowa State University, Ames, Iowa, United States of America

* mercer.97@osu.edu

**Data Availability Statement:** Data and analysis scripts underlying this manuscript has been archived on FigShare. A DOI has been created for the dataset: https://doi.org/10.6084/m9.figshare.12933854.

## Abstract

Responses to drought within a single species may vary based on plant developmental stage, drought severity, and the avoidance or tolerance mechanisms employed. Early drought stress can restrict emergence and seedling growth. Thus, in areas where water availability is limited, rapid germination leading to early plant establishment may be beneficial. Alternatively, germination without sufficient water to support the seedling may lead to early senescence, so reduced germination under low moisture conditions may be adaptive at the level of the population. We studied the germination response to osmotic stress of diverse chile pepper germplasm collected in southern Mexico from varied ecozones, cultivation systems, and of named landraces. Drought stress was simulated using polyethylene glycol solutions. Overall, survival time analysis revealed delayed germination at the 20% concentration of PEG across all ecozones. The effect was most pronounced in the genotypes from hotter, drier ecozones. Additionally, accessions from wetter and cooler ecozones had the fastest rate of germination. Moreover, accessions of the landraces *Costeño Rojo* and *Tusta* germinated more slowly and incompletely if sourced from a drier ecozone than a wetter one, indicating that slower, reduced germination under drought stress may be an adaptive avoidance mechanism. Significant differences were also observed between named landraces, with more domesticated types from intensive cultivation systems nearly always germinating faster than small-fruited backyard- or wild-types, perhaps due to the fact that the smaller-fruited accessions may have undergone less selection. Thus, we conclude that there is evidence of local adaptation to both ecozone of origin and source cultivation system in germination characteristics of diverse chile peppers.

**Funding:** This research was partially funded by Seeds: The OARDC Research Enhancement Competitive Grants Program (Proposal # 2016-056) to KLM, LKM, and LJB and (Proposal #2015-100) to VMB, KLM, and LKM. (oardc.osu.edu/research-resources/grants-contracts) This research, including the original collection trips, was partially funded by the Center for Applied Plant Sciences, Ohio State University. (caps.osu.edu) Salaries and research support were also provided to LKM and KLM by state and federal funds appropriated to the Ohio Agricultural Research and Development Center, Ohio State University: manuscript no. HCS20:09. (oardc.osu.edu). Further salary support was provided by the U.S. Department of Agriculture, Agriculture Research Service. Mention of tradenames or commercial products in this publication is solely for the purpose of providing specific information and does not imply recommendation or endorsement by the U.S. Department of Agriculture. USDA is an equal opportunity provider and employer. The funders had no role in study design, data collection and analysis, decision to publish, or preparation of the manuscript.

**Competing interests:** The authors have declared that no competing interests exist.

# Introduction

Landraces, or traditional varieties, are maintained by farmers through yearly seed saving, allowing them to evolve over time. Landraces may form part of a larger metapopulation covering a landscape with variable abiotic conditions, such as variable water availability [1–3]. As a result, a given landrace population may become locally adapted, likely out-performing non-local populations in the local environment due to particular adaptations selected by the environment [4]. Genotype-by-environment interactions (G×E) [5] indicative of local adaptation have been documented in many cultivated species. In highland and lowland tropical maize (*Zea mays* L.) [6] and inland and coastal barley (*Hordeum vulgare* L.) [7], G×E in final productivity signaled local adaptation. Yet we know less about how the traits at earlier life stages, such as survival at the emergence stage [8] and flowering time [9], may ultimately contribute to the final productivity and fitness variation underlying adaptation to varied environments.

In addition to local adaptation, the process of domestication can also affect ecologically relevant traits in crop plants, influencing their responses to the environment. Traits associated with domestication, including increased seed size, reduced seed shattering, altered growth habit, and decreased seed dormancy [10, 11], can vary among varieties or landraces. Rapid germination, enabled by reduced dormancy, tends to establish a more uniform crop [12, 13], and larger seed size can provide competitive advantages over weeds [14]. Seed structures and components that have shifted with domestication can influence germination [15, 16]. For instance, degree of hull (or pericarp) thickness and openness in various crop-wild hybrids of sunflower (*Helianthus annuus* L.) has been related to variation in germination timing and environmental response [17]. In addition to natural selection, the path and intensity of changes associated with domestication has been affected by farmers working in their respective socio-cultural contexts, e.g., subsistence farm field vs. intensive plantation [18]. Thus, for any crop species, there may be variation among populations for important early life stage characteristics, as well as among plants that have been grown in different cultivation systems.

One abiotic stress that has selected for many adaptations in plants is water deficit. Drought events can vary in duration, magnitude, and severity [19]. A plant's drought resistance, defined as the ability to survive and sometimes grow during periods of water shortage, requires biological mechanisms that protect, sustain, and support the plant during drought events, or to avoid them altogether [20]. These mechanisms may vary depending on the plant species or developmental stage and drought resistance at one stage is not necessarily correlated with resistance at another [21, 22]. Thus, it is important to explore the stage specific drought adaptations of important plant species. In particular, variation in seed germination and dormancy in response to water deficit requires scrutiny.

Drought resistance can be separated into several different strategies: dehydration avoidance, dehydration tolerance, and dehydration escape [23]. For instance, maintaining seeds in an un-germinated state during dry periods can allow them to escape dehydration, although seed dormancy has many other adaptive dimensions (as reviewed in [24]). However, in cultivated species, reductions in dormancy through domestication and further selection may counteract environmental selection on adaptive dormancy behaviors [11, 15]. Thus, it is unclear to what degree domesticated species might express adaptation at the seed stage and how optimal adaptation might vary across environments and production systems.

Chile pepper (*Capsicum* spp.), a Solanaceous crop, possesses significant cultural and economic importance in Mexico. Southern Mexico harbors a large amount of phenotypic diversity and may comprise one of chile's centers of domestication [25, 26]. Within Mexico, chile diversity continues to exist in wild and domesticated populations, to evolve *in situ*, and to experience natural gene flow between wild and domesticated types. Farmers plant traditional

and improved landraces for selling at local markets, but may also cultivate landraces in their home gardens and forage from wild peppers in fields and forests [25]. However, little is known about how domestication level, cultivation systems, ecogeographical structure, and landrace identity organize diversity or affect particular traits, such as those relevant to germination.

Though few studies have investigated germination requirements in *Capsicum*, differences in seed germination between wild and domesticated, and pungent and sweet chile accessions have been noted. Wild chile seeds tend to require light and temperature fluctuations to signal germination, while cultivated chile will germinate readily without these environmental cues [27, 28]. Germination in chile peppers has been found to be affected by fruit chemistry, seed coat thickness, and seed scarification [29]. Seeds with thinner or more porous seed coats (e.g., as a result of scarification) may be able to respond to environmental germination cues more quickly [30]. Furthermore, exogenously applied capsaicin can delay and reduce germination of non-pungent peppers, indicating that capsaicin content alone affects chile germination patterns [31]. However, we are unaware of any work that has been done to assess the effects of a water deficit on chile pepper germination.

The goal of the research presented here was to investigate adaptation to osmotic stress at germination in diverse landraces of chiles collected from different cultivation regimes and ecozones within southern Mexico. We used polyethylene-glycol (PEG) to induce osmotic stress at the seed stage, as has been done in germination studies of other cultivated and wild species [32–40]. In most prior studies, PEG solutions were used to identify cultivars which perform well under both stressed and un-stressed conditions, whereas here we aimed to identify G×E interactions among populations as an indication of local adaptation to drought stress.

We will address the following three objectives using germination curves: 1) To what degree does germination respond to the combination of osmotic stress and environment of origin or cultivation system of origin? 2) Do landraces from the same environment of origin (or cultivation system) share germination responses to osmotic stress? 3) Can we disentangle the effects of ecozone (or cultivation system) on germination responses to osmotic stress from that of differences among landraces? Understanding how germination patterns differ may provide insight into the natural and human-mediated selection pressures that have shaped germination patterns in diverse chile peppers.

## Materials and methods

### Study system and plant material

We collected germplasm of three chile pepper species (*Capsicum annuum* L., *C. fructescens* L., and *C. chinense* Jacq.) from two states of southern Mexico: Oaxaca and Yucatán in 2013 and 2014 (as described in Taitano et al., 2019 [41]). Seed collections were conducted on private and community-held land with direct consent of the land-owner or a community representative. This work did not involve endangered or protected species.

Based on a post-hoc principal component analysis (PCA) of 18 bioclimatic variables and elevation [42] from the collection sites, the collection locations could be separated into five different ecozones (Fig 1). The Western Coast of Oaxaca (W Coast) receives nearly twice as much annual precipitation as the Eastern Coast of Oaxaca (E Coast). The Central Valleys of Oaxaca receive precipitation similar to the East Coast but have cooler temperatures due to their higher elevations. Collections from the Sierra Sur of Oaxaca (i.e., San Baltazar and Santa Lucia) do not cluster bioclimatically with other geographically close localities in the West Coast because they receive higher precipitation in the prevailing edge of a rain shadow. The Yucatán Peninsula experiences temperatures similar to the Oaxacan coast, but with far less annual rainfall. It is important to note that the collections we studied were collected from four distinct

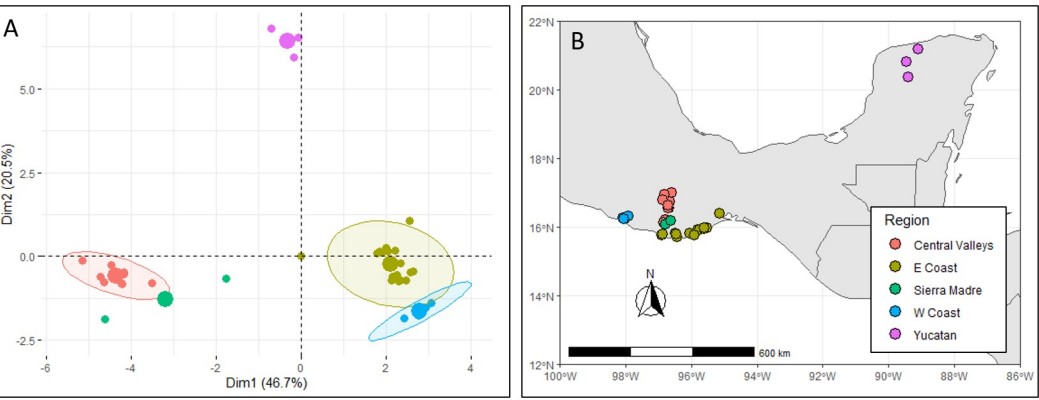

**Fig 1. The five biogeographic ecozones from which we collected chile pepper accessions in Mexico.** Identified on a principal component analysis (PCA) of elevation and bioclimatic (A), and by geographical distribution of accessions within those ecozones (B). This figure was created using data from WorldClim [42] and Natural Earth. The packages 'FactoMineR' [43] and 'raster' [44] were used in the R statistical environment (ver. 3.5; R Development Core Team, 2019).

cultivation systems: forest (wild populations), backyard garden, milpa (a traditional multi-crop production system), and plantation (intensified monocrop production).

For each chile pepper population from which we collected seed (defined here as a set of maternal plants coming from the same named landrace at a given location or farm), we included seeds from one to six randomly chosen individuals (hereafter, accessions) (S1 Fig). We planted these seeds in the greenhouse and grew them under uniform conditions to minimize maternal environmental effects on seed germination in their progeny [45]. One or two plants of each accession were transplanted into in six-liter pots for a total of 190 plants (hereafter, lines) representing 103 accessions, 36 populations, and 22 named landraces. Plants were allowed to grow until 1000 seeds had been harvested. Immature and damaged seeds were sorted out using a light table. Not all lines in the grow-out produced sufficient seeds for experimental use. The resulting seeds (to be used in the germination experiment below) represented 72 accessions of 36 populations and 18 named landraces.

## Experimental design

We designed a germination experiment to assess the response of our diverse chile pepper seeds to varied moisture availability. Specifically, the overall germination pattern was assessed under four levels of osmotic stress simulated with 0, 10, 15, and 20% (w/v) 6000-molecular weight polyethylene glycol (PEG-6000). We divided the experiment into eight runs due to restrictions on space for sample preparation. Each run had an incomplete set of treatment combinations (i.e., PEG level by chile line combinations), and two consecutive runs constituted a complete block. Within each of the four blocks, treatments were randomized with the restriction that all treatment combinations associated with a given line were assigned to the same run. Though four levels of PEG were included in our experiment, statistically significant differences were primarily limited to the comparisons between the 0% PEG water control and 20% concentration of PEG, so the presentation of results will focus there.

Seeds from each line were surface sterilized for 10 minutes with a 0.825% sodium hypochlorite solution (bleach) and rinsed thoroughly in carbon-filtered water. Ten seeds of each line were evenly spaced on blue blotter paper (Anchor Paper Co.; St. Paul, Minnesota) in 100×15-mm petri plates moistened with equal volumes of the prepared solutions. Petri plates were placed in a germinator (Conviron G30; Winnipeg, Manitoba, Canada) for 495 to 542

hours (there was variation among runs) receiving 12 hours of light/dark at 30˚C/20˚C [46]. Seeds remaining ungerminated at the end of the testing period were rinsed with sterile water, placed on new germination paper moistened with water, and scored again after 24 hours. Any newly germinated seeds were considered to have been alive and ungerminated at the end of the treatment. Seeds that continued to remain ungerminated were tested for viability with a 2,3,5-triphenyl tetrazolium chloride (TZ) staining technique [47–49]. To do so, seeds were bisected lengthwise, placed in a 0.075% TZ solution, incubated for three and a half hours at 37˚C, and visually scored. A normal pink-red color indicated a healthy, viable seed; a dark red color indicated an unhealthy, non-viable seed; and no color indicated a dead, non-viable seed.

## Data collected

Thus, each seed was categorized into one of three response variables: germinated, viable (including seeds which germinated in the extended germination window or were scored as viable during the tetrazolium test), or non-viable following germination and viability assays. In the analysis of time-to-event data, germination was given a status code of 1, and failure to germinate was coded as 0 at the final time point. Seeds which germinated or tested positive on a viability test received a viable code of 1, and seeds which did not germinate due to plate contamination or poor seed quality were given a viable code of 0. Plates with fewer than eight viable seeds were omitted from the analysis. A total of 2,342 plates were run, and of these, 1,011 were included in the analysis. The remaining plates represented 131 lines, 18 named landraces, 36 populations and 72 accessions (S1 Table).

## Data analysis

In order to better understand the factors affecting the germination patterns of our diverse chile lines with and without PEG treatments, we compared germination curves using survivorship functions to assess overarching differences.

Time-to-event analysis, also known as survival time analysis, was performed using the packages 'survival' (ver. 2.44) [50, 51] and 'survminer' (ver. 0.4) [52–54] in the R statistical environment (ver. 3.5; R Development Core Team, 2019). In germination studies, a survival function provides the probability that an individual seed will continue to not germinate or to survive (i.e., to remain as a seed) at a given point in time. Since the exact time of germination or failure (i.e., germination means failing to remain ungerminated) cannot be calculated for each seed based on data collected at intervals, the Kaplan-Meier estimator allows us to approach the actual survival function for a population with a sufficient sample size. An important advantage of the Kaplan-Meier estimator is its ability to utilize right-censored data [55]. In other words, it is assumed that ungerminated, viable seeds would germinate at a point past (or, on a time scale, to the right of) the last observation.

We generated Kaplan-Meier estimates of survivor functions for germination of chile seeds separately for each level of individual experimental factors and their combinations. Unfortunately, a limitation of the analysis of these functions is that full models cannot be utilized that test effects of various factors simultaneously or interactions between them. Therefore, we combined levels of multiple factors (e.g. PEG level and ecozone, PEG level and cultivation), as needed, to concurrently assess effects of PEG and genetic factors. In total, we performed six overall comparisons of survival functions by first estimating the survivor functions and then performing pairwise comparisons of all curves. With these survival time analyses, we (1) documented the main effects of PEG on germination and (2) clarified how ecozone of origin and (3) cultivation system affected germination with and without osmotic stress. We then (4) assessed differences among landraces within a given region with and without osmotic stress

for effects on germination and (5) teased apart effects on germination of ecozone of origin from that of landrace. For (5) we subsetted the data by landrace and queried differences among populations of *Tusta* and *Costeño Rojo* landraces collected in multiple ecozones separately with and without osmotic stress. Finally, we (6) determined differences among cultivation system of origin within the Eastern Coast ecozone for germination with and without osmotic stress.

In each case, we used log-rank tests to compare the survival distributions of different levels of each factor. If the survival distribution was significantly different (α was set at 0.05), differences were further dissected by pairwise comparisons and p-values were adjusted for multiple comparisons using the Bonferroni correction.

## Results

### Response to osmotic stress

The PEG treatment significantly reduced germination of the chile seeds relative to the water control (0% PEG) at 15% and 20% concentrations (S2 Fig). There was no statistical difference in the germination curves of the 0% and the 10% PEG treatment (Table 1 in S3 File). While the 15% PEG treatment produced an intermediate response, it was much closer to the water control than the 20% PEG treatment (S2 Fig). Therefore, all further presentation of results from the pairwise contrasts of germination curves and univariate analyses will focus on comparisons of the highest concentration of PEG (20%) and the water control (0% PEG).

PEG affected germination curves specifically by increasing time to germination, increasing $t_{50}$, and reducing final germination (S2 Fig). The time to germination in the water control (0% PEG) was approximately 60 hours, while the time to germination in the 20% PEG treatment was significantly longer (>100 hours) (S2 Fig). The control resulted in approximately 10% of seeds remaining ungerminated, while osmotic stress quadrupled that (approximately 45%) (S2 Fig).

For accessions from each ecozone, we found a qualitatively similar response to the PEG treatment. Overall, the germination curves for each ecozone at 0% PEG were significantly different from the same ecozone at 20% PEG (Table 2 in S3 File). Nevertheless, seeds from the West Coast accessions experienced the smallest effect of PEG, and seeds from the East Coast and Yucatán accessions were most sensitive to PEG levels, with 20% PEG reducing overall germination to less than 50% and increasing the delay to germination (Fig 2).

As in each ecozone, the germination curve for accessions from each cultivation system at 0% PEG was significantly different from the same cultivation system at 20% PEG (Table 3 in S3 File). The accessions studied were collected from four cultivation systems–forest, backyard, milpa, and plantation–which span from unmanaged to highly managed systems. Germination curves of seeds from different cultivation systems were all significantly affected by PEG (Fig 3). With and without osmotic stress, forest seeds germinated most slowly/least and seeds from milpa and plantation systems (the two most intensive production systems) germinated most quickly/completely; seeds from backyard collections fell in between.

When the collected populations were designated according to population type—landrace, letstanding, and wild—we saw patterns related to and partially confounded with cultivation systems. Seeds from wild accessions germinated more slowly/less completely than the letstanding and landrace accessions both with and without osmotic stress (S3 Fig; Table 4 in S3 File).

### Variation among landraces from the same ecozone

In order to investigate if landraces from the same environment of origin share germination responses to osmotic stress, we compared landraces within the same ecozone. We found

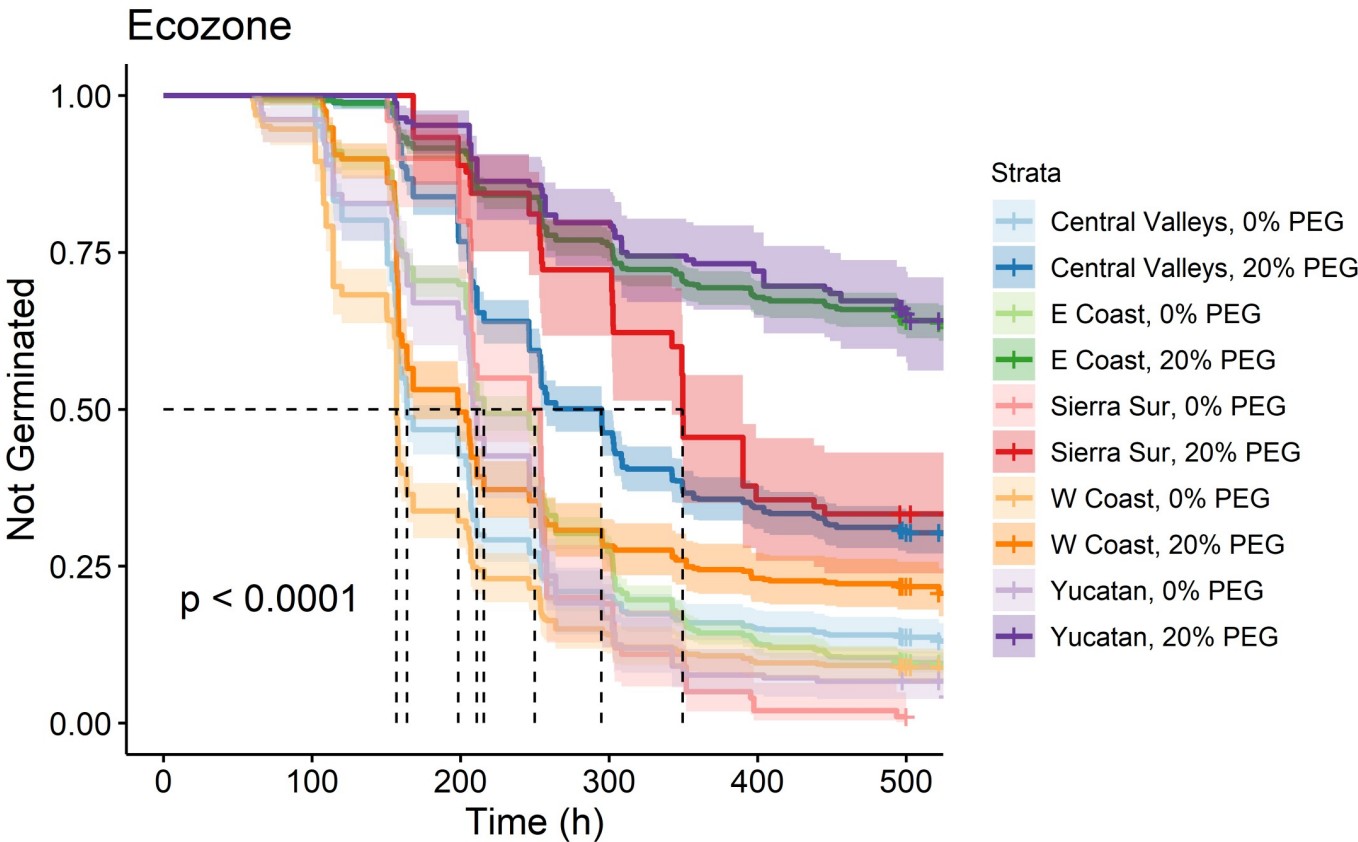

**Fig 2. Kaplan-Meier estimates of survivor functions (and their 95% confidence intervals) for chile pepper accessions originating from different ecozones in southern Mexico.** The p-value from a log-rank test compares survival distributions. The dotted line represents the time to 50% germination ($t_{50}$). Pairwise comparisons of individual curves are presented in Table 2 in S3 File.

significant variation among germination curves from different landraces from the same ecozone with and without osmotic stress (Figs 4A–4D; 5A and 5B). While PEG significantly affected germination curves of most landraces in most regions by significantly slowing their descent and increasing the final proportion of ungerminated seeds, there were exceptions (e.g., see *Costeño Amarillo*, 0% PEG vs *Costeño Amarillo*, 20% PEG from the West Coast; Fig 4A, Table 5 in S3 File). We also saw a mix in whether landraces from a given region were differentiated within a PEG level, with some regions having greater differentiation than others. In the Yucatán, *Paradito*, a small-fruited backyard garden type, had seeds that germinated slower and less completely than those of *Dulce*, a large-fruited landrace grown in milpas, when treated with both 0 and 20% PEG (Fig 4B; Table 6 in S3 File). In the Central Valley, for *Chile de Agua*, *Taviche*, and *Tusta*, landraces, at both 0% and 20% PEG there were no significant differences among their germination curves (Fig 4C; Table 7 in S3 File).

While only one *C. annuum* accession was collected in the Sierra Sur (Fig 4D; Table 8 in S3 File), we collected ten accessions of *C. annuum* from the Eastern Coast, as well as a *C. frutescens* (Fig 5A and 5B; Table 9 in S3 File). Under the 0% PEG treatment, *Gui'ña dani* and *Payaso* were the fastest to germinate. *Solterito* and *Chile de Monte* were the slowest landraces, initiating germination after about 200 hours, at which point *Gui'ña dani* and *Payaso* had already reached 50% germination (Fig 5A). Under 20% PEG, *Costeño Rojo* germinated faster than *Gui'ña dani* or *Payaso*, arriving at $t_{50}$ about 50 hours earlier, although the three landraces had similar final germination (Fig 5B). Interestingly, none of the other seven accessions from the

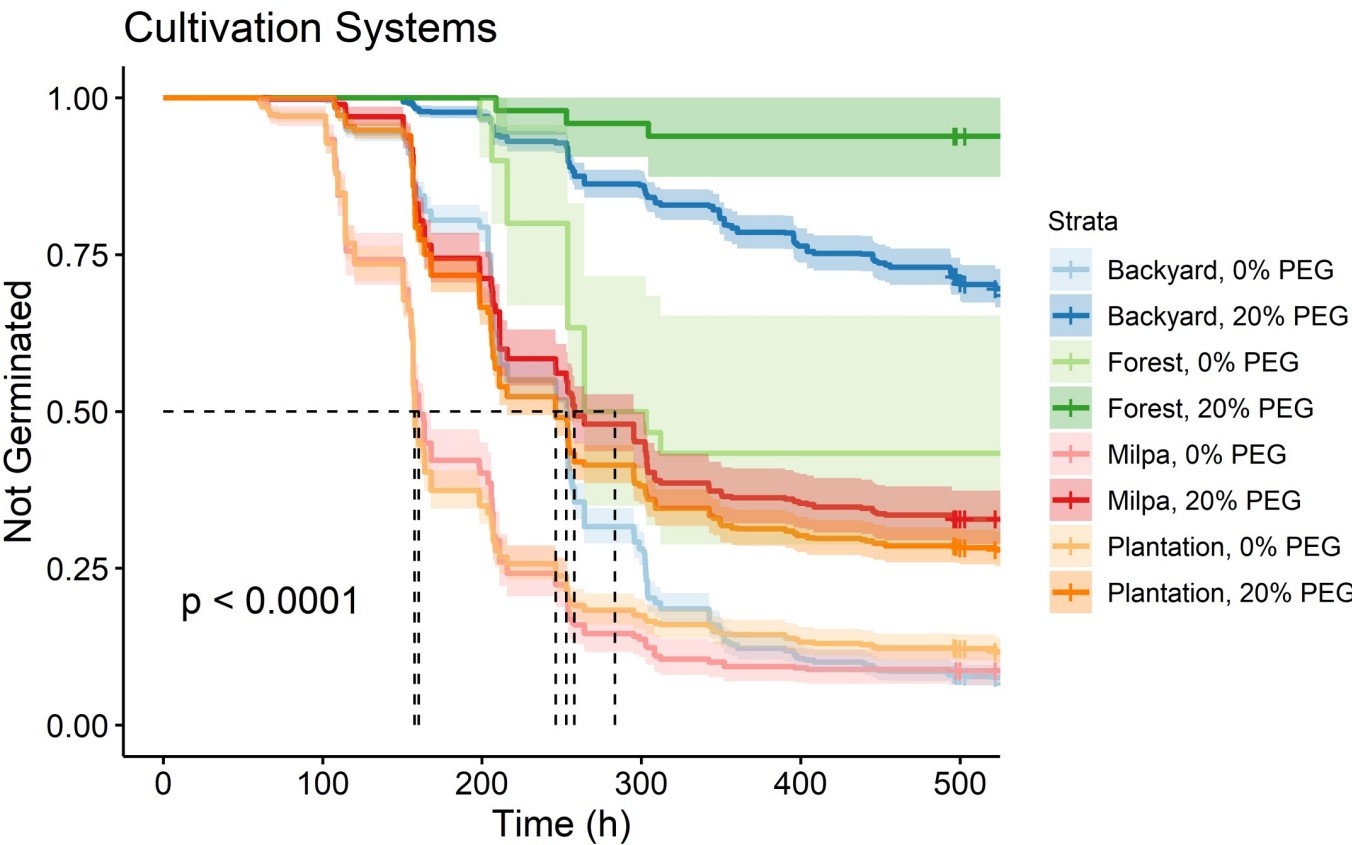

**Fig 3. Kaplan-Meier estimates of survivor functions (with their 95% confidence intervals) for accessions from different cultivation systems of chile pepper in southern Mexico.** The p-value from a log-rank test compares survival distributions. The dotted line represents the time to 50% germination ($t_{50}$). Pairwise comparisons of individual curves are presented in Table 3 in S3 File.

Eastern Coast—all backyard garden landraces—reached 50% germination ($t_{50}$) under the 20% PEG treatment. In fact, *Mareño*, *Mirasol*, and *Solterito* all experienced a delay to first germination of more than 300 hours.

### Variation within cultivation systems from the same ecozone

To disentangle the effects of ecozone and cultivation system, we compared germination curves among cultivation systems from within the same ecozone. This provided a good summary of the variation in germination patterns observed in the Eastern Coast. As in the overall comparison of ecozones, the milpa and plantation accessions (represented by the landraces *Payaso* and *Tusta*, and *Gui'ña dani* and *Costeño Rojo*, respectively) germinated the fastest and were not significantly different under the 0% PEG treatment (Fig 6, Table 10 in S3 File). However, in contrast to what was observed in the overall comparison of cultivation system, the difference between accessions from milpas and plantations is significantly different under the 20% PEG treatment (Figs 3 and 6). As the backyard and forest cultivation systems are represented solely by accessions from the Eastern Coast, their curves remain unchanged relative to Fig 3.

### Variation within landraces sourced from different ecozones

Two landraces in our collections—*Costeño Rojo* and *Tusta*—allow us to compare germination patterns for a given landrace across ecozones to disentangle the effects of ecozone from the

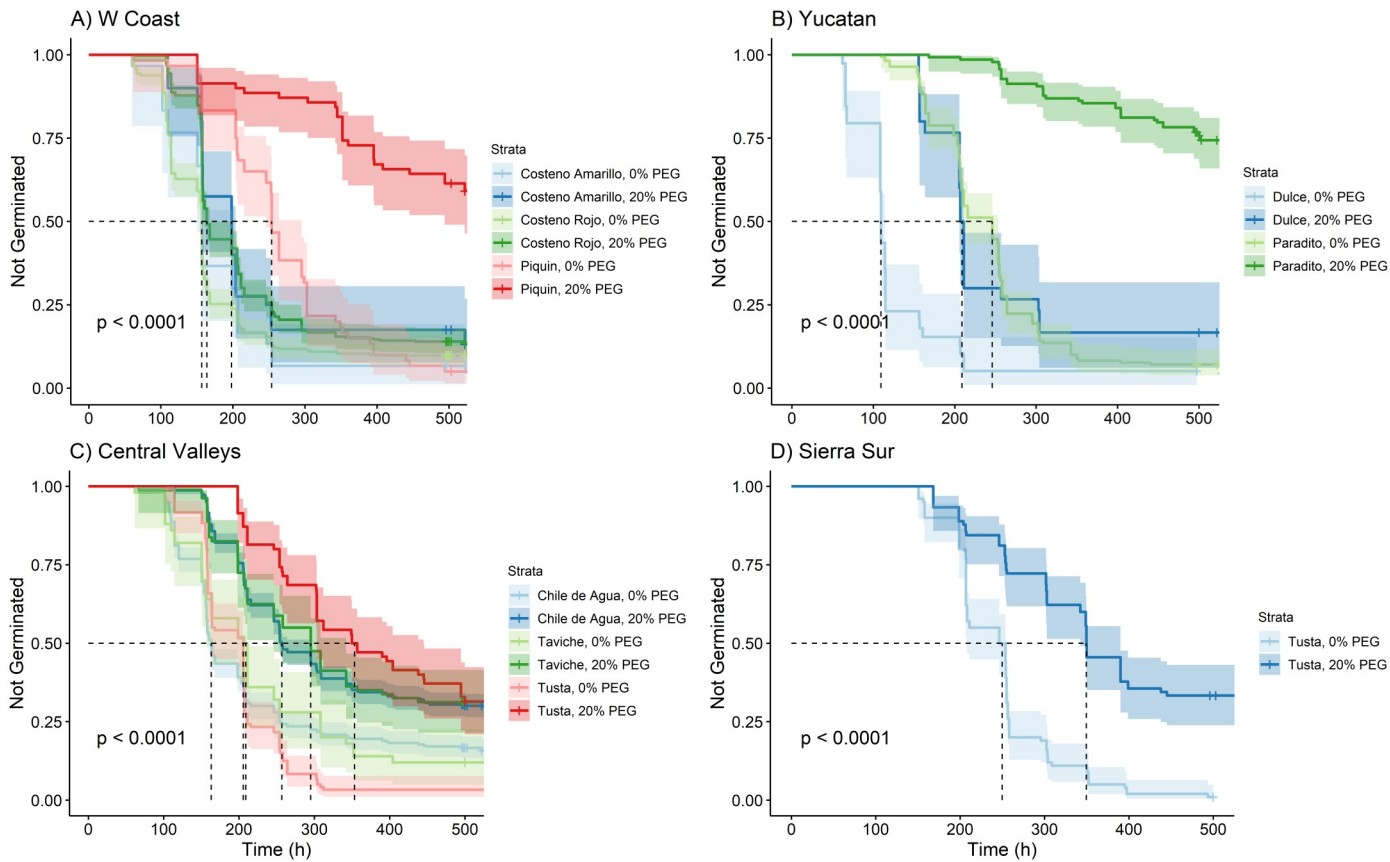

**Fig 4. Kaplan-Meier estimates of survivor functions (and their 95% confidence intervals) for chile pepper originating from each ecozone in southern Mexico.** (A) Western Coast; (B) Yucatan; (C) Central Valleys; (D) Sierra Sur. The p-value from a log-rank test compares survival distributions. The dotted line represents the time to 50% germination ($t_{50}$). Pairwise comparisons of individual curves are presented in Tables 5–8 in S3 File.

reality that the distribution of most landraces is limited to a single ecozone. For *Costeño Rojo*, the germination curves from the Eastern Coast differed in their germination with PEG, with the 20% PEG treatment greatly increasing the proportion of seeds that remained ungerminated (Fig 7A; Table 11 in S3 File). By contrast, those from the Western Coast started out different, but ended similarly (Fig 6A). For *Tusta*, at 0% PEG, seeds from the Sierra Sur and the Eastern Coast did not differ in their germination and germinated slower than those from the Central Valleys (Fig 7B; Table 12 in S3 File), whose seeds reached $t_{50}$ about 50 hours faster. The PEG treatment affected germination for *Tusta* accessions from all three ecozones, but seeds from the Eastern Coast germinated more slowly with PEG than did those from the Central Valleys and Sierra Sur and never reached $t_{50}$. It appears from the case of these two landraces, accessions from different ecozones respond differently to osmotic stress.

## Discussion

In this study, we determined the factors influencing the germination characteristics of diverse chile pepper seed as they responded to PEG-simulated osmotic stress. Based on germination curves, we found that the seeds of landraces that were collected from drier ecozones tended to have slower and less complete germination, especially under osmotic stress. Furthermore, we found that within landraces collected from multiple ecozones (*Tusta* and *Costeño Rojo*), the accessions from the drier Eastern Coast of Oaxaca also tended to have slower and less complete

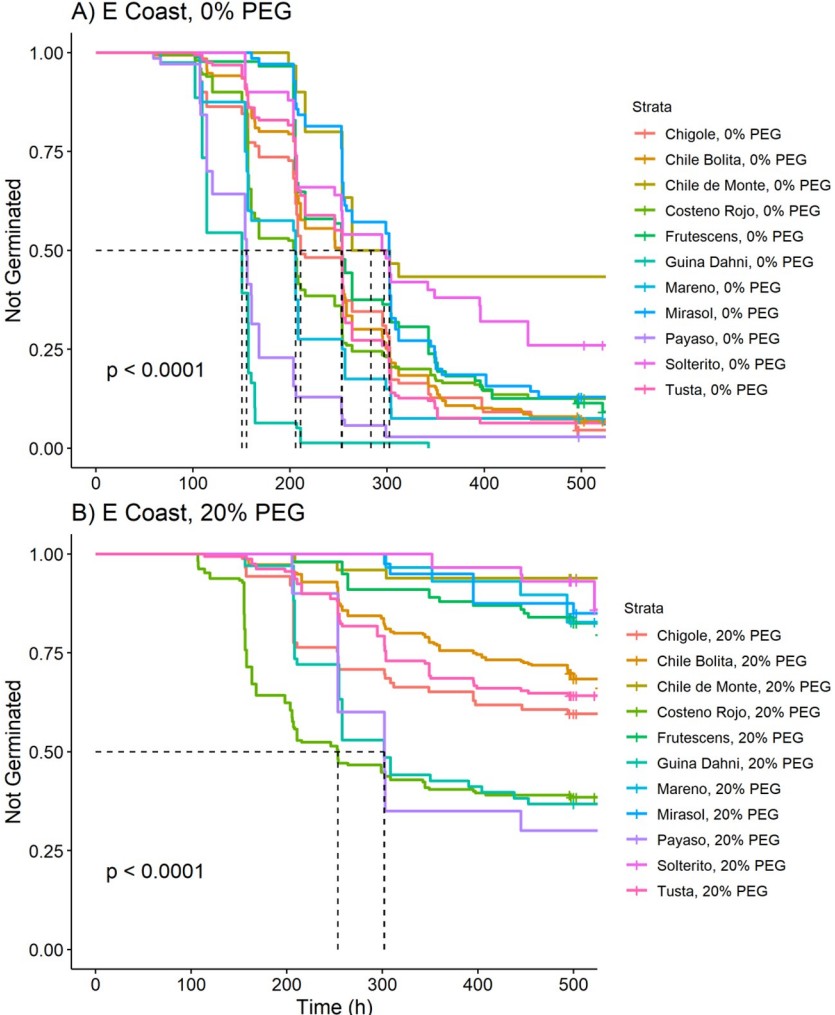

**Fig 5. Kaplan-Meier estimates of survivor functions for chile pepper accessions originating from the Eastern Coast of Oaxaca, Mexico.** (A) Eastern Coast, PEG = 0%; (B) Eastern Coast, PEG = 20%. The p-value from a log-rank test compares survival distributions. The dotted line represents the time to 50% germination ($t_{50}$). Pairwise comparisons of individual curves are presented in Table 9 in S3 File.

germination. Despite this signal of possible adaptation to environment, we found variation within regions among landraces for germination characteristics. Moreover, we found variation in germination patterns to be highly structured by the cultivation system from which accessions were collected. Seeds of chiles collected from forest and backyard cultivation systems demonstrated lower overall germination and a significantly slower germination rate than those collected from milpas and plantations—an effect mirrored by lower germination in wild than in let-standing or landrace chile seeds. Thus, both the environment and the cultivation system from which chiles were collected appear to have influenced germination characteristics and responses to osmotic stress.

## Local adaptation in germination response

While ecozones varied for germination characteristics under each level of PEG, we found that ecozones with comparable environments had similar responses to PEG. For example,

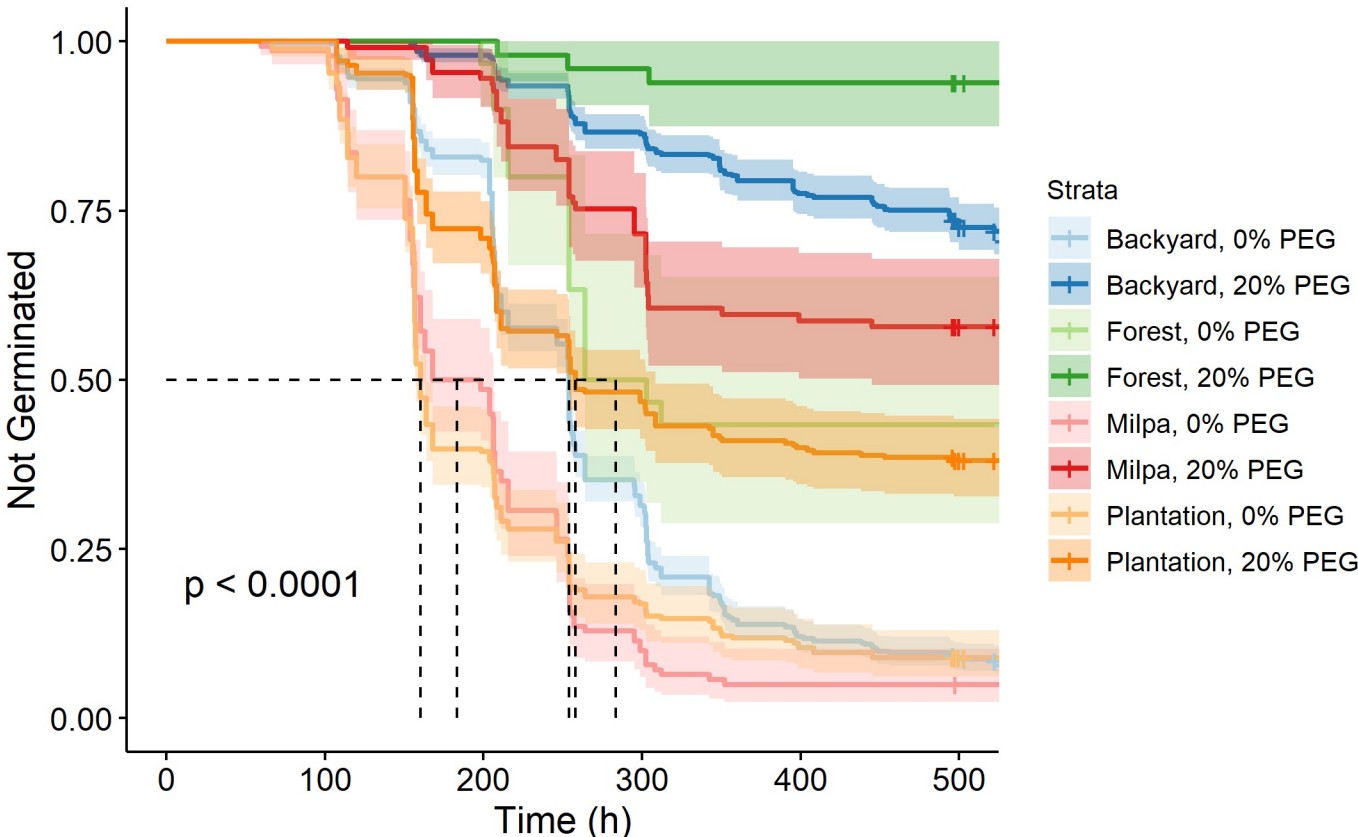

**Fig 6. Kaplan-Meier estimates of survivor functions (and their 95% confidence intervals) for chile pepper accessions originating from the Eastern Coast of Oaxaca, Mexico according to cultivation system of origin.** The p-value from a log-rank test compares survival distributions. The dotted line represents the time to 50% germination ($t_{50}$). Pairwise comparisons of individual curves are presented in Table 10 in S3 File.

accessions from the two drier ecozones, the Yucatán and the Eastern Coast, exhibited similar patterns of germination under osmotic stress. This indicates that they may both possess the same germination adaptations—ones that could be beneficial to survival in dry environments. As discussed by Tewksbury et al. [29], seed coat porosity allows seeds to respond more quickly to germination cues [30]; and early emergence has been found to enhance plant fitness for long-lived perennials such as the wild-type chiles [as reviewed in 56]. Our comparisons of different accessions of the same named landrace sourced from different ecozones made this effect more definitive since accessions with the most similar genetic backgrounds and cultivation histories also showed faster and more complete germination when originating from wetter environments. Despite the complicating factor of human-mediated selection on germination, we do appear to see a consistent pattern of differential adaptation to moisture conditions. Local adaptation to differential moisture (and other environmental pressures) through dormancy has been noted in other species, including *Arabidopsis thaliana* (L.) [57], influencing seedling establishment, survival, and ultimately fecundity (reviewed in [58]).

Specific landraces are often restricted to a geographic region for cultural and/or physiological reasons. Cultural determinants of distribution include landraces being typical of a single indigenous group or community, and thus restricted in their distribution as has been demonstrated in maize [59]. This is the case of *Gui'ña dani* in the Isthmus of Tehuantepec, cultivated almost exclusively by the Didixazá (of the Zapotec languages family). However, landraces may

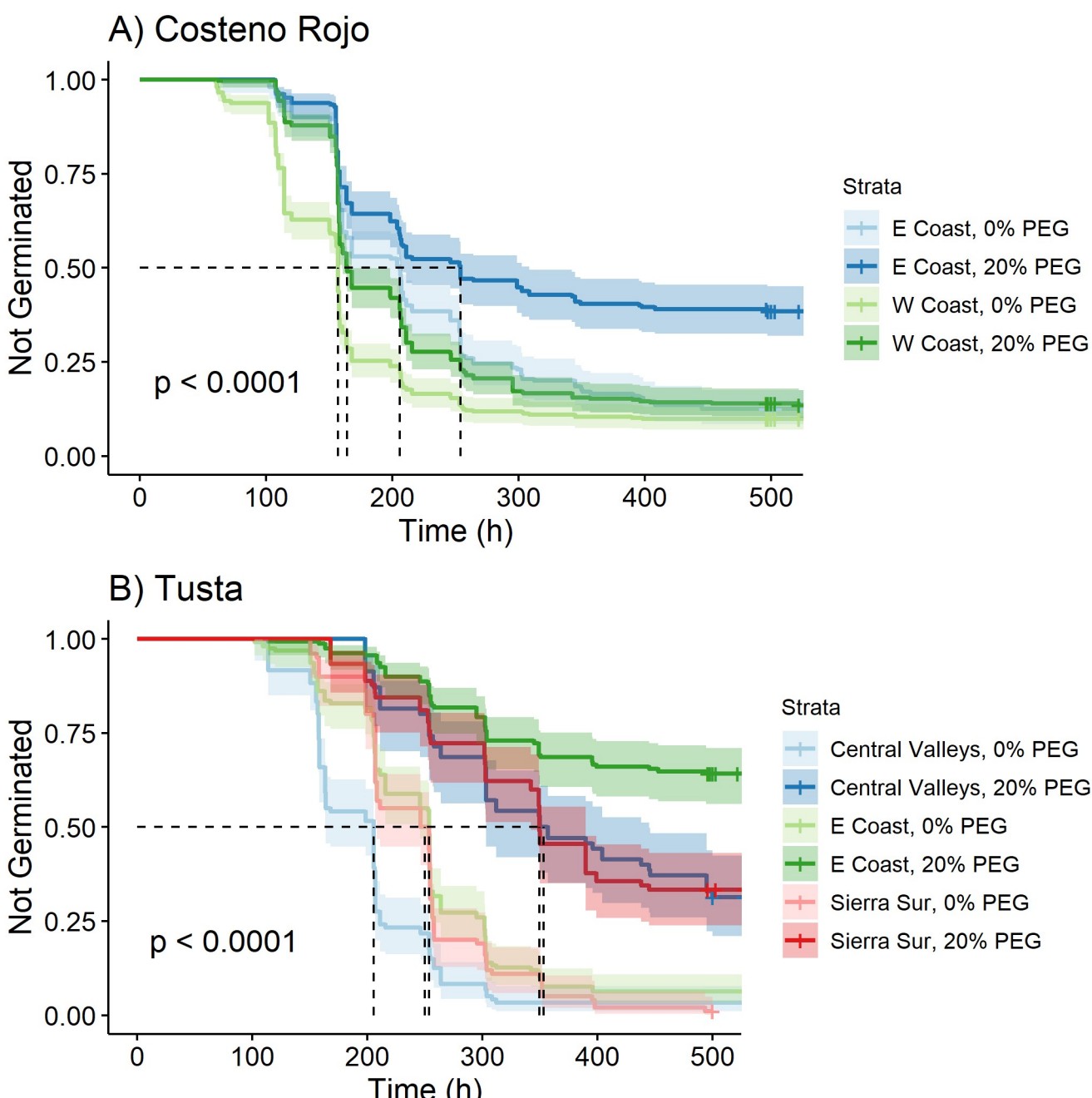

**Fig 7. Kaplan-Meier estimates of survivor functions (and 95% confidence intervals) for chile pepper accessions of the landraces *Costeño Rojo* (A) and *Tusta* (B) collected from different ecozones.** The p-value from a log-rank test compares survival distributions. The dotted line represents the time to 50% germination ($t_{50}$). Pairwise comparisons of individual curves are presented in Tables 11 and 12 in S3 File.

additionally, or alternatively, be restricted by local adaptation to particular pockets of diverse landscapes, as has also been demonstrated in maize in of southern Mexico [9]. To discern local adaptation in this system, we would need to determine the degree to which particular germination or emergence traits influence survival and fruit production in the resulting plants [e.g., 60, 61]

## Cultivation system as a proxy for domestication level and fruit size

Adaptation to cultivation system has primarily been studied at a grosser scale than we do here, by simply comparing wild vs. cultivated ecotypes or related species. In general, our work confirms this literature in that it has found germination to be higher in cultivated than wild types [62, 63] likely due to the intense selection on seed traits during domestication [11]. However, by focusing on cultivation system, we have been able to probe finer differences. For instance, much of the chile production in the Central Valley involves the landraces *Chile de Agua* and *Taviche* and occurs in milpas and plantations (including production in modern greenhouses). Though chiles grown in both milpa and plantation systems are still recognized as landraces, these large-fruited types have been highly selected, especially those grown on plantations, and, in some cases, may be more similar to an improved variety [41]. Backyard landraces constitute an even less strict management system. Thus, by distinguishing milpa and plantation cultivated chile, as well as backyard and wild, we provide a more nuanced view of differences wrought by selection under cultivation.

Cultivation system is also confounded with fruit size. During domestication, fruit size has likely been a main selection criteria for most crops [11, 16]. Furthermore, across seed crops, germination inhibition has been shown to decline and germination rate increases with domestication [15, 64]. For chiles in particular, fruit size can be a key factor for determining domestication level, with the larger fruited, more domesticated types most commonly found in plantations and milpas and smaller fruited, less (or un-) domesticated types found in forests and backyards. Although fruit size was not a component of this study, when looking at the landraces across all ecozones, the factors of domestication level, fruit size, and rate of seed germination can all be seen as indicators of the cultivation system from which the accession was collected. The eight small-fruited landraces (*Chigole*, *Chile Bolita*, *Chile de Monte*, *Mareño*, *Mirasol*, *Payaso*, *Solterito*, and *C. frutescens* spp.; [33]) were mainly collected from backyard gardens (*Payaso* being the only exception), and large-fruited landraces (*Costeño*, *Gui'ña dani*, *Chile de Agua*, and *Tusta*) were typically collected from milpa or plantation systems. (*Tusta* was the exception since it has a medium sized fruit and was collected from backyard gardens in addition to milpas). Large-fruited landraces typically germinated much faster and to a higher total percentage than small-fruited landraces.

The path of a seed from maturation in the fruit to germination is very different in agroecosystems compared to the wild, resulting in different selection pressures acting on seed phenotypes. Seeds of wild chiles are dispersed by frugivorous birds [29, 65], so wild-type fruits are coevolved with the gape of a bird's mouth. When chiles are removed from the wild system for human cultivation and consumption, frugivory no longer plays a role in the dispersal and survival of the offspring (except when seeds are naturally dispersed into backyards where seedlings may not be tended until they are noticed). Instead, rapidity and completeness of germination are valued and selected for. However, in the wild, the effect of frugivory extends beyond dispersal of the seed—it also provides scarification within the bird's digestive system. Furthermore, capsaicin, the compound which gives chiles their pungency, has a costive effect on the digestive tract—increasing the gut-retention time of birds feeding on chile fruits [29]. In this same study [29], *C. annuum* seeds with greater amounts of capsaicin were found to have thicker seed coats, protecting them during passage through the bird's gut. Thicker seed coats would likely contribute to slower imbibition of water into the seed, extending the delay to germination and slowing germination rate in more wild type peppers. The peppers in our study collected from let-standing and wild populations were possibly a result of deposition from frugivorous birds. These let-standing populations (landraces *Payaso* and *Tusta*) showed an interesting G×E response to the PEG treatment. Under the 0% PEG treatment, the

let-standing populations germinated faster and more completely than the landrace populations ($p < 0.001$), but under the 20% PEG treatment there was no significant difference in the response of let-standing and landrace populations (S3 Fig). Thus, even without scarification from frugivory, these intermediates between intensively cultivated landraces and wild chiles might have fast imbibition, but also maintain increased sensitivity to environmental cues.

In addition to deterring mammals, preventing fungal damage, and slowing the digestive tract of birds, capsaicin content has a direct effect on germination. However, capsaicin is not synthesized directly in the seeds or on the seed coat [31, 66, 67], but is instead transferred to the seeds from placental tissue during seed extraction from fruits. Exogenously applied capsaicin has been found to delay and reduce germination of non-pungent peppers, indicating that capsaicin level could be the source of some variation in chile germination patterns [31]. Though capsaicin plays many roles in protecting wild chiles, it is an expensive secondary compound for the plant to make, and capsaicin levels might be expected to decline if not selected by natural systems or farmers [29, 68]. Furthermore, in wild populations, the regulation of seed germination by light, temperature, and soil moisture is an adaptation towards establishing a permanent seed bank and promoting the survival of seeds that germinate [27]. Further work is required to understand the role of capsaicin content and seed coat thickness in the germination of these accessions.

Different germination patterns may be, in part, a result of differences in pepper propagation. In most pepper-producing parts of the world, pepper seeds are not planted directly in the field; they are first grown as seedlings in a greenhouse or outdoor nursery and later transplanted into the field or garden. In the Central Valleys, we witnessed farmers making small nurseries beside their milpas and plantations where they brought in sandier soil from other parts of their property. The sandier soil heats up quickly, providing a warm seed bed for germination but necessitating more vigilant monitoring of moisture post germination and possibly unlinking climate from early life stages. A survey of farmer propagation techniques may provide further insight into the germination responses presented here.

## Conclusions

The research presented here has shown significant variation in germination response to drought-mimicking seed treatments among seeds collected from different ecozones of origin and cultivation systems in southern Mexico. Though we expected to, and did, see a strong effect of ecozone of origin organizing germination response, we also found that the effect of cultivation system (an indicator of landrace improvement and selection intensity) has a strong effect. This study is the first of its kind to disentangle these two, often confounded, factors and thus helps us better understand the factors shaping phenotypic diversity in our crops.

## Supporting information

**S1 Fig. Representation of sampling strategy.** Seeds or fruits were collected from 28 different communities in Southern Mexico. In each community, several farms or locations were visited (total of 59). Seed from one or more landraces were collected at each location. A population is made up of one or more maternal plants of the same landrace from the same location. An accession represents seeds sampled from an individual plant in the field. One or two lines in the greenhouse were developed from each accession.
(TIF)

**S2 Fig. Kaplan-Meier estimates of survivor functions for seeds of chile pepper treated with multiple PEG levels.** The p-value from a log-rank test compares survival distributions. The

dotted line represents the time to 50% germination ($t_{50}$). Pairwise comparisons of individual curves are presented in Table 1 in S3 File.
(TIF)

**S3 Fig. Kaplan-Meier estimates of survivor functions for accessions from different population types of *Capsicum* spp. in southern Mexico exposed to differing osmotic stress simulated by 0% and 20% PEG solutions.** The p-value from a log-rank test compares survival distributions. The dotted line represents the time to 50% germination ($t_{50}$). Pairwise comparisons of individual curves are presented in Table 4 in S3 File.
(TIF)

**S1 Table. The distribution of populations, accessions and lines across landraces studied.**
(PDF)

**S1 File. R markdown output from PCA and map (Fig 1).**
(HTML)

**S2 File. R markdown output from survival time analysis.**
(HTML)

**S3 File. Pairwise comparison tables for all survival curve figures.** P-values within each table were adjusted for multiple comparisons with a Bonferroni correction.
(PDF)

## Acknowledgments

We would like to acknowledge Amanda Gutek, Matthew Willman, Katherine D'Amico, Jacob Hite (deceased), and John Brett for their help with the laboratory preparation. We thank Pablo Jourdan, Peter Curtis, Alex Lindsey, and Emmy Regnier for reading earlier versions of this manuscript. We also thank Brian Pace, Rachel Capouya, Esther van der Knaap, Nathan Taitano, Luis E. Eguiarte, Catarino Perales Segovia, Alejandra Moreno Letelier, Jose Luna, Jose Carillo, and Salvador Montes for their help with the collections and assisting in the transfer of seeds to OSU. Finally, we thank the many chile growers who graciously allowed us into their homes and shared with us the fruits of their labor. This study is founded upon generations of their work.

## Author Contributions

**Conceptualization:** Vivian M. Bernau, Leah K. McHale, Kristin L. Mercer.

**Formal analysis:** Vivian M. Bernau.

**Funding acquisition:** Leah K. McHale, Kristin L. Mercer.

**Investigation:** Vivian M. Bernau.

**Project administration:** Leah K. McHale, Kristin L. Mercer.

**Resources:** Vivian M. Bernau, Lev Jardón Barbolla, Leah K. McHale, Kristin L. Mercer.

**Supervision:** Leah K. McHale, Kristin L. Mercer.

**Visualization:** Vivian M. Bernau.

**Writing – original draft:** Vivian M. Bernau.

**Writing – review & editing:** Vivian M. Bernau, Lev Jardón Barbolla, Leah K. McHale, Kristin L. Mercer.

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
