## [Decision Letter · Decision Letter 0]

12 Aug 2020

PONE-D-20-19642

Germination response of diverse wild and landrace chile peppers (Capsicum spp.) under drought stress simulated with polyethylene glycol

PLOS ONE

Dear Dr. Mercer,

Thank you for submitting your manuscript to PLOS ONE. After careful consideration, we feel that it has merit but does not fully meet PLOS ONE’s publication criteria as it currently stands. Therefore, we invite you to submit a revised version of the manuscript that addresses the points raised during the review process.

I have now received one review report. Based on the report and my opinion, the manuscript can be published in Plos One after necessary minor revisions.There are confusions on the seed germination parameters studied. Please clarify these.There are some typos and differences in the use of names of different accessions.Please make these corrections and submit a revised version for further evaluation.

We look forward to receiving your revised manuscript.

Kind regards,

Shahid Farooq, Ph.D.

Academic Editor

PLOS ONE

Journal Requirements:

3. Please include a caption for figure 5.

4. We note that Figure 1 in your submission contain map images which may be copyrighted. All PLOS content is published under the Creative Commons Attribution License (CC BY 4.0), which means that the manuscript, images, and Supporting Information files will be freely available online, and any third party is permitted to access, download, copy, distribute, and use these materials in any way, even commercially, with proper attribution. For these reasons, we cannot publish previously copyrighted maps or satellite images created using proprietary data, such as Google software (Google Maps, Street View, and Earth). For more information, see our copyright guidelines: http://journals.plos.org/plosone/s/licenses-and-copyright.

4.1.    You may seek permission from the original copyright holder of Figure 1 to publish the content specifically under the CC BY 4.0 license. 

4.2.    If you are unable to obtain permission from the original copyright holder to publish these figures under the CC BY 4.0 license or if the copyright holder’s requirements are incompatible with the CC BY 4.0 license, please either i) remove the figure or ii) supply a replacement figure that complies with the CC BY 4.0 license. Please check copyright information on all replacement figures and update the figure caption with source information. If applicable, please specify in the figure caption text when a figure is similar but not identical to the original image and is therefore for illustrative purposes only.

Reviewers' comments:

Reviewer's Responses to Questions

**Comments to the Author**

1. Is the manuscript technically sound, and do the data support the conclusions?

Reviewer #1: Yes

2. Has the statistical analysis been performed appropriately and rigorously? 

Reviewer #1: Yes

3. Have the authors made all data underlying the findings in their manuscript fully available?

Reviewer #1: Yes

4. Is the manuscript presented in an intelligible fashion and written in standard English?

Reviewer #1: Yes

5. Review Comments to the Author

Reviewer #1: The work presented interesting data about pepper culture under stressful condition.

- Please avoid les colored area around the curve. In some figures, it is difficult to differentiate, colored lines are sufficient.

- References must be added for the parameters.

- Line 288 : « while PEG significantly affected germination curves of most landraces in most regions by significantly slowing their descent and reducing the final proportion of ungerminated seeds »

Comment : It is rather ‘increasing the final proportion…..’, please verify.

- Line 306 : « While only one C. annuum accession was collected in the Sierra Sur (Fig 5A and Fig 5B;Table 8 in S3 File),… »

Comment : ….(Fig. 4D ; …..)

- Line 308, 322 : Please use the same name for the accession : Gui’ña dani or Guina Dahni or Guiña Dahni.

- For median germination time, noted here : t50:

Comment: to note, there is a difference between the mean or median germination time, usually noted (MGT), and the time taken for cumulative germination to reach 50% of its maximum, usually noted (t50) (as it is described by Gammoudi et al. (2020), Efficiency of pepper seed invigoration through hydrogen peroxide priming to improve in vitro salt and drought stress tolerance).

In the current manuscript, it is neither this nor that, here, it is simply the time to reach the percentage 50%, so, for that it was noted : « none of the other seven accessions from the Eastern Coast—all backyard garden landraces—reached 50% germination (t50) under the 20% PEG treatment », whereas, in fact, for each curve, we have a MGT and t50. Please verify.

6. PLOS authors have the option to publish the peer review history of their article (what does this mean?). If published, this will include your full peer review and any attached files.

Reviewer #1: **Yes: **Najet Gammoudi

---

## [Author Response · Author response to Decision Letter 0]

8 Oct 2020

We are grateful for the suggestions you provided. We feel that by addressing them we have improved the manuscript.

---

## [Editor Report · Decision Letter 1]

12 Oct 2020

Germination response of diverse wild and landrace chile peppers (Capsicum spp.) under drought stress simulated with polyethylene glycol

PONE-D-20-19642R1

Dear Dr. Mercer,

We’re pleased to inform you that your manuscript has been judged scientifically suitable for publication and will be formally accepted for publication once it meets all outstanding technical requirements.

Kind regards,

Shahid Farooq, Ph.D.

Academic Editor

PLOS ONE

Additional Editor Comments (optional):

I have gone through the revised manuscript. All of the comments were addressed by the authors. Therefore, the current version can be accepted for publication.
---

## [Editor Report · Acceptance letter]

5 Nov 2020

PONE-D-20-19642R1 

Germination response of diverse wild and landrace chile peppers (*Capsicum* spp.) under drought stress simulated with polyethylene glycol 

Dear Dr. Mercer:

I'm pleased to inform you that your manuscript has been deemed suitable for publication in PLOS ONE. Congratulations! Your manuscript is now with our production department. 

Kind regards, 

on behalf of

Dr. Shahid Farooq 

Academic Editor

PLOS ONE